# Induced Human Regulatory T Cells Express the Glucagon-like Peptide-1 Receptor

**DOI:** 10.3390/cells11162587

**Published:** 2022-08-19

**Authors:** Anna K. O. Rode, Terkild Brink Buus, Veronika Mraz, Fatima Abdul Hassan Al-Jaberi, Daniel Villalba Lopez, Shayne L. Ford, Stephanie Hennen, Ina Primon Eliasen, Ib Vestergaard Klewe, Leila Gharehdaghi, Adrian Dragan, Mette M. Rosenkilde, Anders Woetmann, Lone Skov, Niels Ødum, Charlotte M. Bonefeld, Martin Kongsbak-Wismann, Carsten Geisler

**Affiliations:** 1The LEO Foundation Skin Immunology Research Center, Department of Immunology and Microbiology, Faculty of Health and Medical Sciences, University of Copenhagen, DK-2200 Copenhagen, Denmark; 2Grünenthal GmbH, Zieglerstr. 6, 52078 Aachen, Germany; 3Novo Nordisk A/S, DK-2760 Måløv, Denmark; 4Lundbeck, DK-2500 Copenhagen, Denmark; 5Laboratory for Molecular Pharmacology, Department of Biomedical Sciences, University of Copenhagen, DK-2200 Copenhagen, Denmark; 6Department of Dermatology and Allergy, Herlev and Gentofte Hospital, University of Copenhagen, DK-2900 Copenhagen, Denmark

**Keywords:** human, CD4^+^ T cells, GLP-1R expression

## Abstract

The glucagon-like peptide-1 receptor (GLP-1R) plays a key role in metabolism and is an important therapeutic target in diabetes and obesity. Recent studies in experimental animals have shown that certain subsets of T cells express functional GLP-1R, indicating an immune regulatory role of GLP-1. In contrast, less is known about the expression and function of the GLP-1R in human T cells. Here, we provide evidence that activated human T cells express GLP-1R. The expressed GLP-1R was functional, as stimulation with a GLP-1R agonist triggered an increase in intracellular cAMP, which was abrogated by a GLP-1R antagonist. Analysis of CD4^+^ T cells activated under T helper (Th) 1, Th2, Th17 and regulatory T (Treg) cell differentiation conditions indicated that GLP-1R expression was most pronounced in induced Treg (iTreg) cells. Through multimodal single-cell CITE- and TCR-sequencing, we detected GLP-1R expression in 29–34% of the FoxP3^+^CD25^+^CD127^-^ iTreg cells. GLP-1R^+^ cells showed no difference in their TCR-gene usage nor CDR3 lengths. Finally, we demonstrated the presence of GLP-1R^+^CD4^+^ T cells in skin from patients with allergic contact dermatitis. Taken together, the present data demonstrate that T cell activation triggers the expression of functional GLP-1R in human CD4^+^ T cells. Given the high induction of GLP-1R in human iTreg cells, we hypothesize that GLP-1R^+^ iTreg cells play a key role in the anti-inflammatory effects ascribed to GLP-1R agonists in humans.

## 1. Introduction

It is well established that obesity and type 2 diabetes mellitus are associated with an increased risk of inflammatory diseases such as cardiovascular disease [1], nephropathy and psoriasis [2,3]. Over the last 15 years, glucagon-like peptide-1 receptor agonists (GLP-1RA) have proven successful for the treatment of both obesity and diabetes [4]. Glucagon-like peptide-1 (GLP-1) is produced by enteroendocrine L cells in the gut in response to contact with nutrients [5] and acts via the GLP-1 receptor (GLP-1R). The GLP-1R is a G protein-coupled receptor (GPCR) and activation of the GLP-1R leads to an increase in intracellular cAMP and Ca^++^ and promotes ERK1/2 signaling [5]. GLP-1 stimulates insulin secretion and inhibits glucagon release [4,5]. Furthermore, GLP-1 decreases gastric emptying and suppresses appetite and food intake. Several studies have indicated that GLP-1RA, in addition to their insulinotropic and anorexigenic effects, have anti-inflammatory effects. Thus, GLP-1RA treatment results in a significant 10% relative risk reduction in cardiovascular disease [6,7], reduces development of diabetic kidney disease [8] and improves psoriasis [9,10,11,12]. In addition, a meta-analysis has demonstrated that treatment with GLP-1RA causes a significant reduction in C-reactive protein, further supporting the existence of an anti-inflammatory effect of GLP-1RA [13].

The anti-inflammatory potential of GLP-1RA could be due to a direct effect of GLP-1RA on immune cells, an indirect effect caused by weight loss, or a combination of the two. A direct effect of GLP-1RA on immune cells would require that they express the GLP-1R. Regulatory T (Treg) cells constitute a critical part of immune cells that controls immune responses and resolves inflammation [14]. The anti-inflammatory roles of Treg cells are most obviously seen in subjects with the immune dysregulation, polyendocrinopathy, enteropathy, or X-linked (IPEX) syndrome caused by mutations in the transcription factor forkhead box protein P3 (FoxP3) [15,16]. FoxP3 is required for Treg cell maintenance, and IPEX patients lack Treg cells and suffer from severe multi-organ autoimmunity that leads to death within their first years of life if untreated [17]. Treg cells are divided in natural Treg (nTreg) cells that develop in the thymus, and peripheral or induced Treg (iTreg) cells that develop from naïve CD4^+^ T cells in the periphery. Apart from being CD4^+^CD25^+^FoxP3^+^, Treg cells are highly heterogeneous. Thus, 9 and 22 distinct subpopulations of human Treg cells have been identified by single-cell transcriptomic analysis and mass cytometry, respectively [18,19].

GLP-1R expression has been described in mouse T cells, especially in intestinal intraepithelial lymphocytes [20,21]. However, only a few studies have focused on GLP-1R expression in human T cells. One study found that GLP-1RA affected chemotaxis of CD4^+^ T cells and demonstrated that CD4^+^ T cells express the GLP-1R by Western blot analysis [22]. Another study found that invariant natural killer T (iNKT) cells express GLP-1R mRNA, and that GLP-1RA inhibited the secretion of the cytokines interferon γ (IFNγ) and interleukin-4 (IL-4) in activated iNKT cells [9]. Finally, a recent study found that peripheral blood mononuclear cells (PBMC) express GLP-1R mRNA and protein, and that treatment with a GLP-1RA increased the production of IFNγ [23]. Although somewhat opposing results were obtained, these studies collectively suggest that some human T cells express the GLP-1R. The aim of the present study was to characterize GLP-1R expression in different subsets of human CD4^+^ T cells.

## 2. Material and Methods

### 2.1. Purification and Culturing of T Cells

Peripheral blood mononuclear cells (PBMC) were isolated by Lymphoprep (1114547, Axis-Shield, Oslo, Norway) density gradient centrifugation using SepMate^TM^ tubes (85450, Stemcell Technologies, Grenoble, France) from healthy donors after obtaining informed, written consent in accordance with the Declarations of Helsinki principles for research involving human objects as previously described [24]. The study was approved by The Committees of Biomedical Research Ethics for the Capital Region in Denmark (H-16033682). Naïve CD4^+^ T cells were isolated and cultured as previously described [25] using the EasySep Human Naïve CD4^+^ T cell Enrichment Kit (19155, Stemcell Technologies, Vancouver, Canada). The purified naïve T cells were cultured in serum-free X-VIVO 15 medium (BE02-060F, Lonza, Verviers, Belgium) at 37 °C, 5% CO_2_ at a cell concentration of 1 × 10^6^ cells/mL in flat-bottomed 24-well tissue culture plates (142475, Nunc, Roskilde, Denmark) and activated with Dynabeads Human T-Activator CD3/CD28 beads (111.31D, Life Technologies, Grand Island, NY, USA) at a bead to cell ratio of 2:5 for up to 5 days.

In Th polarization experiments, naïve CD4^+^ T cells were cultured and stimulated in the presence of recombinant human IL-12 (10 ng/mL, 219-IL) and anti-IL-4 antibodies (4 µg/mL, MAB204) for Th1 polarization; recombinant human IL-4 (10 ng/mL, 204-IL) and anti-IFNγ antibodies (4 µg/mL MAB285) for Th2 polarization; recombinant human TGFβ (5 ng/mL, 240-B), IL-6 (20 ng/mL, 206-IL), IL-21 (30 ng/mL, 8879-IL), IL-23 (30 ng/mL, 1290-IL), anti-IL-4 antibodies (4 µg/mL), and anti-IFNγ antibodies (4 µg/mL) for Th17 polarization; and TGFβ (5 ng/mL), IL-2 (100 IU/mL, Proleukin, Clinigen, London, UK), retinoic acid (RA, 100 nM, R2625, Sigma Aldrich, Søborg, Denmark), and rapamycin (Rapa, 100 ng/mL, 553210, Calbiochem, Merck, Darmstadt, Germany) for iTreg cells. All polarization reagents were from R&D Systems unless otherwise stated. In some experiments 100 nM 25-hydroxyvitamin D_3_ (25(OH)D_3_, BML-DM-100-0001, Enzo Life Sciences, Inc., Ann Arbor, MI, USA) was added to the medium at the start of activation.

### 2.2. RT-qPCR

mRNA was purified and reverse transcribed into cDNA as previously described [25,26]. mRNA levels for various targets were quantified using RT-qPCR by mixing cDNA with TaqMan^®^ Universal Master Mix II with UNG (4440038, Applied Biosystems, Waltham, MA, USA) and target primers. The following primers were used: GLP-1R (Hs00157705_m1 and alternative primer Hs01006332_m1 targeting another region of the human GLP-1R gene), FoxP3 (Hs01085834_m1), PDCD1 (PD-1) (Hs01550088_m1), CD274 (PD-L1) (Hs01125301_m1), IL2RA (CD25) (Hs00907777_m1), TGFβ (Hs00998133_m1) and GAPDH (Hs02758991_g1), all from Applied Biosystems.

### 2.3. Adenylyl Cyclase Activation FlashPlate^®^ Assay

The cAMP accumulation assay was carried out using the Adenylyl Cyclase Activation FlashPlate^®^ Assay from Perkin Elmer (SMP004) according to manufacturer’s instructions. The assay was carried out in a scintillation covered 96-well FlashPlate based on the principle that intracellular cAMP accumulation after ligand stimulation of cells competes with a fixed concentration of radiolabeled cAMP for binding to an immobilized anti-cAMP antibody. Exendin-4, forskolin, prostaglandin E1 (PGE1) and exendin 9–39 (made in-house at Novo Nordisk) were diluted in PBS containing 0.2% pluronic F-68 (24040032, Gibco, Waltham, MA, USA) and added directly to the wells of the FlashPlate. Subsequently, 5 × 10^5^ T cells per 50 μL stimulation buffer were added to the wells in duplicates. The stimulation buffer contained the phosphodiesterase inhibitor 3-isobutyl-1-methylxanthine (IBMX) to prevent cAMP degradation. Naïve CD4^+^ T cells were either left unstimulated or cultured like above in Th1 or Th2 medium with or without 100 nM 25(OH)D_3_. The plates were covered in tin foil, shaken for 5 min, and incubated for 25 min at room temperature. A detection mix containing the radiolabeled cAMP was added to the wells and the plates were shaken for 30 min more and left at room temperature for 2–3 h before being analyzed in a TopCount^®^NXT^TM^ Scintillation and Luminescence Counter. A cAMP standard curve was included in each plate and was plotted in GraphPad Prism^®^ for each experiment. Concentration–response curves for the exendin-4 induced cAMP accumulation were fitted to a three-parameter logistic model. All FlashPlate^®^ assays and data analyses were done at the Incretin Biology group at Novo Nordisk A/S, Måløv.

### 2.4. Flow Cytometric Analysis

For flow cytometric analysis of T cells, the activation beads were removed using a magnet and the cells were washed and re-suspended in FACS buffer and stained with anti-CD25-BV786 (M-A251) (BD Bioscience, Herlev, Denmark) and Fixable Viability Dye (eFluor780, eBioscience, Waltham, MA, USA) for 30 min at 4 °C in the dark. Cells were permeabilized and fixed using the eBioscience FoxP3 Transcription Factor Staining Buffer Set (00-5523-00) according to manufacturer’s instructions and stained with anti-FoxP3-PE (236A/E7, BD Bioscience, Herlev, Denmark) for 30 min at room temperature in the dark. Samples were run on a five-laser BD LSRFortessa flow cytometer (BD Bioscience, Herlev, Denmark) and analyzed with FlowJo v10 software (Treestar, BD Bioscience, Herlev, Denmark).

### 2.5. Single-Cell RNA Sequencing, CITE- and scTCR-seq

Cellular indexing of transcriptomes and epitopes by sequencing (CITE-seq) was conducted using single-cell immune profiling with feature barcoding (10X Genomics) reagents following the manufacturer’s instructions with some modifications. Cells were multiplexed using cell hashing antibodies targeting β2-microglobulin and CD298 according to the published ECCITE-seq and cell hashing protocols [27,28]. In short, iTreg cells were differentiated from naïve CD4^+^ T cells from three healthy donors as described above. Two × 10^5^ cells from each donor were tagged with cell hashing antibodies with unique barcodes for each donor before being thoroughly washed and pooled. Pooled cells were stained with a panel of oligo-conjugated antibodies including antibodies targeting CD25, CD26 and CD127 (302649, 302722, 351356 BioLegend, San Diego, CA, USA) and washed as previously described [29]. After washing, cells were counted and loaded in a single well of a Chip K and run on the 10X Chromium Controller. Individual libraries for gene expression, TCRαβ, surface antibodies and cell multiplexing (hashtags) were prepared following the manufacturer’s instructions as well as the published ECCITE-seq protocol using dual indexing. Libraries were sequenced together with other runs on a NovaSeq S1 flow cell. Th1/Th17 cells were prepared from a single donor following an analogous protocol, and CITE- and scTCR-seq were conducted following the same approach as described above.

### 2.6. Enrichment of Single-Cell GLP-1R Transcripts

GLP-1R and FOXP3 transcripts were amplified from 50 ng of cDNA using a two-step nested PCR inspired by the genotyping of transcripts protocol [30]. The first PCR used the generic sample indexing forward primer for 10X Chromium (SI-PCR: AATGATACGGCGACCACCGAGATCTACACTCTTTCCCTACACGACGC*T*C) and an outer reverse primer targeting the GLP-1R transcript containing (PCR1_GLP-1R: AGCAAGTGAGAAGCATCGTGTCGCACTGGCGTCGGTATTCTCG; PCR1_FOXP3: AGCAAGTGAGAAGCATCGTGTCCCAGCAGGTCTGAGGCTTTGG) and was amplified for 14 cycles of 98 °C for 20 s, 58 °C for 30 s and 72 °C for 20 s. Products were cleaned up and size-selected by SPRI bead selection (0.45× supernatant, 0.6× supernatant, and 1.4× bead fraction). The second PCR used the SI-PCR forward primer and a nested reverse primer targeting the previous GLP-1R amplicon as well as an overhang containing the PCR handle for the Illumina TruSeq Small RPI-x primer used for indexing (PCR2_GLP-1R: CACCCGAGAATTCCAATCCCGAGCAGCAGCAG*C; PCR2_FOXP3: CACCCGAGAATTCCAGGGTTGGGCATCGGGTC*C) and was amplified for 16 (FOXP3) or 20 (GLP-1R) cycles of 98 °C for 20 s, 58 °C for 30 s and 72 °C for 20 s. Products were cleaned up and size-selected by double-sided (0.7× and 1.6×) SPRI bead selection. Amplicons we quantified using the Qubit4 with dsDNA high-sensitivity kits and pooled at similar concentrations. To assure comparability, identical volumes of amplicons were pooled from the Treg and Th1/17 runs. A finished sequencing library was generated by an indexing PCR using SI-PCR forward primer and an RPI-x reverse-indexing primer (RPI-x: CAAGCAGAAGACGGCATACGAGATXXXXXXXXXXGTGACTGGAGTTCCTTGGCACCCGAGAATTCCA) for 7 cycles of 98 °C for 20 s, 54 °C for 30 s and 72 °C for 20 s and followed by a final 1.5X SPRI bead selection.

### 2.7. Analysis of Single-Cell Data

Gene expression and TCRαβ libraries were processed using Cell Ranger (v6.0.1; 10X Genomics). GLP-1R-enriched libraries were quantified using Kallisto-Bustools [31]. Hashtags and surface antibody libraries were quantified with Kallisto-Bustools using the KITE workflow. Following balancing of the hashtag signal using upsampling, cell barcodes were de-multiplexed using the Hashsolo algorithm [32]. Viable cells were filtered based on having at least 500 detected genes and less than 10% of their total unique molecular identifiers (UMIs) stemming from mitochondrial transcripts. The majority of cell multiplets were removed during demultiplexing. SingleCellExperiment objects from the two runs containing iTreg (D1-D3) and Th1/Th17, respectively, were preprocessed separately and then combined by simple merging (cbind). Gene expression was normalized using the scran: logNormCounts. TCR-segment genes were removed from the variable feature list before dimensional reduction and clustering. Cells were clustered and visualized using the first 14 principal components. Cell cycle state was inferred using Seurat:CellCycleScoring. CITE-seq counts were de-noised and normalized using the ‘dsb’ R package utilizing signal from empty droplets to reduce ambient noise [33].

### 2.8. PBMC Proliferation Assay

For proliferation assays, freshly purified PBMC were stained with CellTrace Violet (CTV, C34557, Thermo Fisher) according to the manufacturer’s instructions. In short, PBMC were washed and re-suspended at 1 × 10^6^ cells/mL in PBS prior to staining for 20 min at 37 °C 5% CO_2_ with 5 µM CTV. A five-time excess of warm X-VIVO 15 medium supplemented with 1% FBS was added, and cell mixtures were allowed to rest for 5 min to remove free dyes. The CTV-labeled cells were pelleted and re-suspended in warm X-VIVO 15 medium at 5 × 10^6^ cells/mL and cultured at 0.25 × 10^6^ cells in round-bottomed 96-well plates in the presence of 100 ng/mL anti-CD3 antibodies (OKT3, 16-0037-81, eBioscience, Waltham, MA, USA) and the indicated concentrations of liraglutide (Victoza, A10BJ02, Novo Nordisk, Bagsværd, Denmark) and forskolin (F6886, Sigma Aldrich, Søborg, Denmark). A CTV-stained sample was left unstimulated to determine the gating of undivided cells. After five days of culture, cells were pelleted, and supernatant was saved for ELISA. Cells were washed with FACS buffer (2% FBS and 0.2% Na-Azide in PBS), incubated with human Fc blocker (564219) and stained with anti-CD3-BUV395 (UCHT1), anti-CD8-FITC (RPA-T8), anti-CD4-APC (RPA-T4), and Fixable Viability Dye (eFluor780) (eBioscience Waltham, MA, USA) for 30 min at 4 °C in the dark. Samples were run on a five-laser BD LSRFortessa flow cytometer (BD Bioscience) and the data were analyzed with FlowJo v10.

### 2.9. ELISA

IFNγ concentrations in the supernatants were determined by ELISA according to the manufacturer’s protocol (Ready-Set-Go; eBioscience Waltham, MA, USA).

### 2.10. Stable Transfection of HEK293 Cells with hGLP-1R

HEK293 cells stably expressing human GLP-1R were generated using the calcium phosphate transfection method [34]. One day prior to the transfection, two T75 flasks were seeded with 2 × 10^6^ cells: one for transfection and one for “death control”. At 24 h after transfection, the cells were released using 1.5 mL 1 × Trypsin-EDTA and a single cell suspension was created in 4.5 mL culture medium (DMEM 1% GlutaMAX supplemented with 10% FBS, 1% penicillin (180 U/mL)/streptomycin (45 µg/mL)). Subsequently, the cells were seeded in five peel-off flasks using the dilution ratios 1/25, 1/50, 1/100, 1/250, and 1/500 in selection medium (culture medium supplemented with 0.4 mg/mL G418) in order to allow single cells to grow into colonies. At the same time, the non-transfected cells (the “death control”) were cultured in a new flask (1/5 cell density ratio) using the selection medium. The cells were monitored using microscopy, and the selection medium was exchanged every 48 h. After 8 days, all the cells in the “death control” flask were dead, while the transfected cells were still growing, except in the 1/500 flask. On day 14, the 1/250 flask had distinct colonies emerging from single cells and 10 of the colonies were transferred to T25 flasks using cloning rings. The selection medium was refreshed every 4–5 days until approximately 60% confluence was reached. The expression of the receptor in each clone was assessed using the DiscoverX cAMP accumulation assay [35].

### 2.11. Immunofluorescence Microscopy on HEK293 Cells and Skin Samples

The wild-type and the hGLP-1R transfected HEK293 cells were grown on coverslips in 24-well plates in a concentration of 6.7 × 10^4^ cells/well for two days. The coverslips were pre-coated with poly-L-lysine (1 µg/mL, diluted 1:1000 in MiliQ water) for 1 h at room temperature and then washed with PBS 3 times for 30 min. Coverslips with cells were removed from the wells and briefly washed with PBS. Cells were fixed in ice-cold acetone–methanol solution (diluted 1:1) for 5 min. Afterwards, coverslips with cells were dried out and washed 3 times in PBS for 5 min. Cells were blocked in 10% goat serum (S26, Sigma-Aldrich, Søborg, Denmark) in PBST (0.1% Tween20 in PBS) for 30 min at room temperature and subsequently stained with recombinant rabbit anti-human GLP-1R monoclonal antibodies (EPR23507-57, Abcam) diluted 1:100 at 4 °C overnight. Secondary staining was performed with goat anti-rabbit IgG Alexa Flour 555 (A27039, Thermo Fisher Scientific, Waltham, MA, USA) diluted 1:500 for 1 h at room temperature. Cells were counterstained with DAPI (62248, Thermo Fisher Scientific, Waltham, MA, USA) diluted 1:1000 for 5 min, mounted with ProLong Diamond Anti-fade mountant (P36982, Life Technologies, Nærum, Denmark) and cured overnight at room temperature. Coverslips with cells were washed 3 times for 5 min in PBST between each staining. Samples were visualized with a Plan-Apochromat 63x numerical aperture 1.4 oil objective (Zeiss) on Zeiss LSM710. The system was controlled by Zen Lite, Zen Black version.

Nine individuals with previous positive patch test reactions to nickel (2+) were enrolled after informed, written consent had been obtained. The study was approved by The Committees of Biomedical Research Ethics for the Capital Region in Denmark (H-15017309). The individuals were patch tested with petrolatum or 5% nickel in petrolatum. Skin biopsies (4 mm) were taken from upper inner arm 72 h after the patch test. Formalin-fixed paraffin-embedded skin biopsies were cut into sections of 4 µm, de-paraffinized in xylene and rehydrated through graded alcohol followed by antigen retrieval at pH 9.0 in a microwave oven twice for 2 min at 120 W and twice for 6 min at 80 W. Sections were blocked in 5% goat serum (S26, Sigma-Aldrich, Søborg, Denmark) for 1 h at room temperature and subsequently stained with mouse anti-human CD4 monoclonal antibodies (14-0049-80, Thermo Fisher Scientific, Waltham, MA, USA) and recombinant rabbit anti-human GLP-1R monoclonal antibodies (EPR23507-57, Abcam) both diluted 1:100 overnight at 4 °C. Secondary staining was performed with goat anti-mouse mAbs Alexa Flour 647 (A21235, Thermo Fisher Scientific, Waltham, MA, USA) diluted 1:500 and goat anti-rabbit mAbs Alexa Flour 555 (A27039, Thermo Fisher Scientific, Waltham, MA, USA) diluted 1:2000 for 2 h at room temperature. Sections were counterstained with DAPI diluted 1:1000 (62248, Thermo Fisher Scientific, Waltham, MA, USA) for 10 min, mounted with ProLong Diamond Antifade mountant (P36982, Life Technologies, Nærum, Denmark) and cured overnight at room temperature. Sections were washed 3 times for 5 min in PBS between each staining. Samples were visualized with a Plan-Apochromat 63x numerical aperture 1.4 oil objective (Zeiss) on a Zeiss Axio Observer Z1 that includes 1x Yokogawa (CSU-X1) spinning disc confocal; 405-, 561-, and 639-nm lasers; and an Orca Fusion camera. The system was controlled by Zen Lite, Zen Blue version.

### 2.12. Statistical Analysis

Statistical analyses were performed as indicated in the figure legends. Graphpad Prism software was used for all statistical analyses.

## 3. Results

### 3.1. Activated CD4^+^ T Cells Express Functional GLP-1R

To investigate whether CD4^+^ T cells express the GLP-1R, we activated naïve CD4^+^ T cells for 0, 24, 48 and 72 h and measured their GLP-1R mRNA levels by RT-qPCR. We found that naïve CD4^+^ T cells expressed low levels of the GLP-1R; however, activation resulted in a rapid, approximately 40-fold up-regulation of the GLP-1R (Figure 1A). To determine whether T-cell differentiation affected GLP-1R expression, we next stimulated naïve CD4^+^ T cells in Th1 or Th2 polarizing medium. Furthermore, we investigated the effect of vitamin D (25(OH)D_3_), as vitamin D is reported to be an important determinant in the differentiation of CD4^+^ T cells [36,37,38]. We found that activation resulted in a 30- and 50-fold up-regulation of the GLP-1R expression in CD4^+^ T cells activated in Th1 and Th2 polarizing medium, respectively, suggesting that T-cell differentiation might affect GLP-1R expression (Figure 1B). Interestingly, the addition of vitamin D increased expression of the GLP-1R in both CD4^+^ T cells activated in Th1 (5-fold) and Th2 (9-fold) polarizing medium compared to cells activated in the absence of vitamin D (Figure 1B). As for naïve CD4^+^ T cells, non-stimulated CD8^+^ T cells, CD4^+^ memory T cells and nTreg cells did not express the GLP-1R, but GLP-1R expression could be detected in these cells after activation (data not shown). We did not further investigate GLP-1R expression in CD8^+^ T cells, CD4^+^ memory T cells and nTreg cells in this study but focused on the expression of the GLP-1R in naïve CD4^+^ T cells activated in vitro.

The GLP-1R is a class B1 GPCR, and binding of GLP-1 to the GLP-1R causes signaling through a stimulatory G protein that activates adenylyl cyclase (Gαs). The adenylyl cyclase converts ATP into cAMP, thereby increasing the intracellular concentration of cAMP [5]. In order to determine whether activated human CD4^+^ T cells express a functional GLP-1R, we measured the ability of CD4^+^ T cells activated under Th2 conditions in the presence of vitamin D to produce cAMP upon GLP-1R stimulation. We found that treatment of the cells with the GLP-1RA exendin-4 induced a significant increase in the cAMP concentration (Figure 1C). To determine the EC_50_ values of exendin-4 for CD4^+^ T cells activated under various conditions, we activated naïve CD4^+^ T cells in Th1 or Th2 polarizing medium in the absence or presence of 25(OH)D_3_. Subsequently, we treated the cells with increasing concentrations of exendin-4 for 30 min and measured the intracellular cAMP concentration. In line with their very low GLP-1R mRNA expression, naïve CD4^+^ T cells did not respond to exendin-4 (Figure 1D,E). In contrast, concentration–response curves were obtained in most of the experiments with CD4^+^ T cells activated under Th1 and Th2 conditions in both the presence and absence of 25(OH)D_3_ (Figure 1D,E). We found that the EC_50_ value for CD4^+^ T cells activated under Th2 conditions in the absence and presence of vitamin D and for CD4^+^ T cells activated under Th1 conditions in the presence of vitamin D were significantly lower than the EC_50_ value for CD4^+^ T cells activated under Th1 conditions in the absence of vitamin D (Figure 1F). The EC_50_ values in the pM range for the T cells were comparable with the EC_50_ value obtained in exendin-4 stimulated fibroblasts transfected with the human GLP-1R [39]. Although CD4^+^ T cells activated under Th2 conditions in the absence of vitamin D apparently expressed less GLP-1R than CD4^+^ T cells activated under Th1 and Th2 conditions in the presence of vitamin D (Figure 1B), there were no statistically significant differences between their EC_50_ values (Figure 1F). To evaluate the receptor specificity of the exendin-4 response further, we treated activated CD4^+^ T cells with exendin-4 in the presence or absence of a 100-fold higher concentration of the GLP-1R antagonist exendin 9–39 [39]. Exendin 9–39 significantly inhibited the exendin-4-induced intracellular cAMP accumulation (Appendix A), supporting that the observed cAMP accumulation after exendin-4 treatment is mediated via GLP-1R activation. We confirmed that exendin 9–39 specifically inhibited GLP-1R signaling, as it did not affect cAMP accumulation triggered by prostaglandin E1 (PGE1) or forskolin stimulation (Appendix A).

Taken together, these data indicated that activated human CD4^+^ T cells express a functional GLP-1R. The relatively low GLP-1R expression levels combined with the low EC_50_ values suggested that only a subset of the activated cells expressed the GLP-1R.

### 3.2. GLP-1R Expression Directly Correlates with the Percentage of CD25^+^FoxP3^+^ T Cells

To identify the subset of activated human CD4^+^ T cells with the strongest expression of the GLP-1R, we activated naïve CD4^+^ T cells for five days in non-supplemented (Th0) medium or Th1-, Th2-, Th17- or Treg cell-polarizing media in the absence or presence of 25(OH)D_3_ and measured their GLP-1R mRNA levels by RT-qPCR. We found that CD4^+^ T cells activated under Treg cell conditions expressed most GLP-1R of the subsets tested in both the absence and presence of vitamin D (Figure 2A). Compared to CD4^+^ T cells activated in Th0 medium, CD4^+^ T cells activated under Treg cell conditions expressed approximately 100 to 150-fold more GLP-1R. As observed in the previous experiments with CD4^+^ T cells activated under Th0, Th1 and Th2 conditions, we found that vitamin D up-regulated GLP-1R expression in CD4^+^ T cells activated under Th0, Th1, Th2 and Th17 conditions. However, vitamin D did not significantly affect GLP-1R expression in CD4^+^ T cells activated under Treg cell conditions (Figure 2A).

Treg cells are characterized as being CD4^+^CD25^+^FoxP3^+^ T cells. We found that CD4^+^ T cells activated under Th0, Th1, Th2 and Th17 conditions contained 2–9% CD25^+^FoxP3^+^ cells, while cells activated under Treg cell conditions contained approximately 40% CD25^+^FoxP3^+^ cells (Figure 2B,C; for the gating strategy please see Appendix A). This suggested that CD25^+^FoxP3^+^ cells might represent the CD4^+^ T cell subset that most strongly expressed the GLP-1R. In line with this, linear regression analyses indicated that GLP-1R expression levels directly correlated with the percentage of CD25^+^FoxP3^+^ T cells in any T cell population activated both in the absence or the presence of vitamin D_3_ (Figure 2D,E). From this, we hypothesized that CD25^+^FoxP3^+^ T cells are the major GLP-1R-expressing CD4^+^ T cell subset.

### 3.3. The Combination of TGFβ, Rapamycin and Retinoic Acid Induces Expression of FoxP3 and GLP-1R in Parallel in T Cells

It has previously been shown that TGFβ in combination with IL-2 is able to induce FoxP3 expression in naïve CD4^+^ T cells [40], but more recent studies have found that rapamycin and retinoic acid (RA) also play important roles in Treg cell induction [41,42,43,44]. To investigate whether there is a correlation between the factors that induce FoxP3 and GLP-1R expression in T cells, we activated naïve CD4^+^ T cells in the presence of various combinations of previously identified FoxP3-inducing factors. We found that TGFβ and rapamycin were able to up-regulate the expression of FoxP3 when given separately, and that the combination of TGFβ, rapamycin and RA induced maximal expression of FoxP3, which was 22-fold higher than FoxP3 expression in cells activated in the absence of any of the components (Figure 3A). The addition of vitamin D and/or IL-2 to the mixture of TGFβ, rapamycin and RA did not further enhance the expression of FoxP3. Interestingly, we found that the combination of TGFβ, rapamycin and RA also induced maximal expression of the GLP-1R, which was approximately 90-fold higher than GLP-1R expression in cells activated in the absence of any of the components (Figure 3B). Furthermore, as seen for FoxP3, addition of vitamin D and/or IL-2 to the mixture of TGFβ, rapamycin and RA did not further enhance the expression of the GLP-1R. Combined with the data shown in Figure 2, these data support that CD4^+^CD25^+^FoxP3^+^ iTreg cells or a subpopulation of these cells are the main GLP-1R-expressing CD4^+^ T cells.

### 3.4. Approximately 30% of iTreg Cells Express the GLP-1R

To determine the proportion of iTreg cells expressing the GLP-1R, we subsequently analyzed iTreg cells differentiated from naïve CD4^+^ T cells from three healthy donors (D1–D3) using single-cell cellular indexing of transcriptomes and epitopes by sequencing (CITE-seq) combined with scTCR-seq. For comparison, we also included Th1 and Th17 cells differentiated from naïve CD4^+^ T cells (Figure 4A). iTregs showed high FOXP3 and IRF4 expression (Figure 4B and Appendix A). Compared to Th1 and Th17 cells, iTreg cells showed high expression of IL-2Rα (CD25) and low expression of IL-7Rα (CD127) and DPP4 (CD26) both at the mRNA and protein level, confirming the CD4^+^ iTreg phenotype (Figure 4C and Appendix A). Consistent with our RT-qPCR data, we found cDNA reads aligning to the exonic regions of the *GLP1R* gene consistent with these cells expressing processed GLP-1R transcripts (Figure 4D). As expected from a lowly expressed gene and the sparsity of single-cell RNA-seq data, relatively few unique transcripts were recovered as measured by unique molecular identifiers (UMI). To improve detection, we enriched for GLP-1R and FOXP3 transcripts from the total mRNA using a two-step nested PCR approach. This drastically improved detection of GLP-1R from 55–71 to 2517–2971 unique transcripts from the three donors (38–54-fold increase in detection; Appendix A). This marked increase in detection sensitivity revealed that 29–34% of the iTreg cells expressed GLP-1R transcripts as compared to 4.8% and 5.3% of Th1 and Th17 cells, respectively (Figure 4E). GLP-1R expression did not appear to be restricted to distinct subpopulations but was expressed across the entire iTreg cell population (Figure 4F). We did notice enrichment of GLP-1R expressing cells within cells that co-expressed FOXP3, confirming the association of the GLP-1R with the iTreg cell phenotype (Figure 4G). By comparing the T cell receptor (TCR) sequences between GLP-1R^+^ and GLP-1R^-^ iTreg cells, we did not find any clear differences in their complementarity-determining region (CDR) 3 length nor in their TCR V-segment usage (Appendix A).

### 3.5. GLP-1RA Inhibit Proliferation of CD4^+^ and CD8^+^ T Cells and Increase PD-1 and PD-L1 Expression in iTreg Cells

Previous studies have indicated that GLP-1RA inhibits the proliferation and cytokine production of T cells in mice [45] and the IFNγ production of activated human PBMC [23]. In an attempt to identify the functional roles of GLP-1R signaling in T cells in addition to the increase in cAMP, we activated PBMC from healthy donors with anti-CD3 mAb in the absence or presence of increasing concentrations of the GLP-1RA liraglutide for 5 days and measured proliferation indexes, live cells and IFNγ production. As a control, we included the general cAMP inducer forskolin. We found that liraglutide inhibited proliferation of both CD4^+^ and CD8^+^ T cells without affecting the viability of the cells or the production of IFNγ (Figure 5A–D and Appendix A). These experiments did not allow us to conclude whether GLP-1RA have a direct anti-proliferative effect on GLP-1R-expressing CD4^+^ and CD8^+^ T cells or whether the effect was mediated via iTreg cells, but they supported that GLP-1RA affect T cell responses. To investigate whether GLP-1RA have a direct effect on iTreg cells, we activated naïve CD4^+^ T cells for five days in Treg cell-polarizing medium. Subsequently, the cells were washed and activated in non-supplemented medium in the absence or presence of exendin-4. After 24 h, selected gene expression levels were measured by RT-qPCR. Interestingly, we found that exendin-4 significantly increased the expression levels of programmed death 1 (PD-1) and programmed death-ligand 1 (PD-L1) (Figure 5E,F), whereas it did not affect expression of TGFβ, CD25 or FoxP3 (Appendix A).

### 3.6. GLP-1R^+^CD4^+^ T Cells Are Found in the Skin

Finally, to exclude that GLP-1R expression in T cells was just an in vitro phenomenon, we tested whether GLP-1R^+^CD4^+^ T cells could be detected in skin from patients with allergic contact dermatitis to nickel. Biopsies of skin treated with vehicle or with nickel were analyzed by immunofluorescence microscopy for CD4 and GLP-1R expression. We found several CD4^+^ T cells in the dermis of both non-lesional and lesional skin that co-expressed the GLP-1R (Figure 6). We confirmed the specificity of the GLP-1R antibody by immunofluorescence microscopy on HEK293 cells transfected with GLP-1R and on wild-type HEK293 cells that do not express the GLP-1R (Appendix A).

## 4. Discussion

In this study, we demonstrate that a sub-population of activated human T cells, and especially iTreg cells, express the GLP-1R, and that GLP-1R^+^CD4^+^ T cells are found in the dermis of both non-lesional and lesional skin from patients with allergic contact dermatitis. The GLP-1R expressed by T cells was functional, as stimulation with GLP-1RA triggered an increase in intracellular cAMP, which was specifically attenuated by the GLP-1R antagonist exendin 9–39. To our knowledge, this is the first study that demonstrates the expression of functional GLP-1R in human iTreg cells. In line with this, two previous studies have demonstrated functional GLP-1R expression in human iNKT cells [9] and in mouse intra-epithelial T cells [20], where treatment with GLP-1RA induced increases in the intracellular cAMP concentrations.

In T cells, we found EC_50_ values for the GLP-1RA exendin-4 in the pM level, which is comparable to EC_50_ values obtained for exendin-4 in fibroblasts transfected with the human GLP-1R [39]. However, the overall increase in cAMP levels in the T cells after GLP-1R stimulation was not as substantial as the response to the cAMP-inducers PGE1 and forskolin. This suggested that only a subset of the T cells expressed the GLP-1R. Vitamin D increased the frequency of FoxP3^+^ T cells and GLP-1R expression in parallel in Th1 and Th2 cells. However, the EC_50_ values for exendin-4 in Th2 cells treated with vitamin D was not significantly lower than the EC_50_ values for untreated Th2 cells as might first be expected. This suggested that vitamin D did not increase the GLP-1R expression level in a given cell but rather increased the number of cells expressing the GLP-1R. In accordance, we found the strongest expression of the GLP-1R in CD4^+^ T cells activated under Treg cell conditions and that GLP-1R expression in a population of T cells correlated with the frequency of CD25^+^FoxP3^+^ T cells. This suggested that CD25^+^FoxP3^+^ iTreg cells might be the main CD4^+^ T cell subset that expressed the GLP-1R expression. This was further supported by the observation that a combination of factors promoting FoxP3 expression promoted GLP-1R expression in parallel. Conclusively, by multimodal single-cell CITE-sequencing we demonstrated that a large subset (approximately 30%) of FoxP3^+^CD25^+^CD127^-^ iTreg cells expressed GLP-1R transcripts. It has been reported that FoxP3 can be transiently expressed in activated human T cells with no immunosuppressive capacity [46]. As we found that GLP-1R expression correlated with the frequency of FoxP3^+^ T cells, it could be suggested that GLP-1R, like FoxP3, could be a transient activation marker. However, our CITE-seq analyses demonstrated that approximately 32% of the FoxP3^+^ iTreg cells expressed the GLP-1R, whereas only 5% of the FoxP3^+^ Th1 and Th17 cells expressed the GLP-1R. This indicated that GLP-1R is not just an activation marker but that it is differential expressed on various activated T cells.

A previous study indicated that resting, non-activated human CD4^+^ T cells express GLP-1R, as stimulation with GLP-1RA affected the chemotaxis of CD4^+^ T cells [22]. This appears to be in contrast to our study, as we could not detect GLP-1R expression in non-activated T cells, and the CD4^+^ cells used in the chemotaxis study were not activated. However, as we studied purified naïve CD4^+^ T cells, and the chemotaxis study used total CD4^+^ T cells treated with chemokines; this might explain the different results. A search in existing datasets in the EMBL-EBI Expression Atlas “Gene expression across species and biological conditions” [47] was in accordance with our results, as no evidence of constitutive expression of the GLP-1R in non-activated naïve T cells, memory T cells or Treg cells was found. Together with our observations, this indicates that GLP-1R expression in T cells requires activation of the cells, and that the GLP-1R expression is transient. We found that GLP-1R activation did not affect the survival or the production of IFNγ in PBMC from healthy donors activated with anti-CD3 mAb. In contrast, a previous study found that GLP-1R activation increased the production of IFNγ in PBMC activated with phytohemagglutinin [23]. However, these PBMC were obtained from patients with chronic obstructive pulmonary disease, and the effect of GLP-1RA on PBMC from healthy donors was not reported. Furthermore, we found that high concentrations of GLP-1RA inhibited cell division of both CD4^+^ and CD8^+^ T cells. We could not conclude whether this was a direct effect of GLP-1RA on a subset of activated CD4^+^ and CD8^+^ T cells that expressed the GLP-1R or if it was an indirect effect mediated via GLP-1R^+^ iTreg cells. However, we found that the GLP-1RA exendin-4 increased the expression of PD-1 and PD-L1 in iTreg cells. An increase in PD-1 and PD-L1 expression might increase the immunosuppressive ability of human iTreg cells as previously described in mice [48,49,50]. The effect of GLP-1RA on T cell proliferation has, to our knowledge, not been studied in humans, but results from studies performed in mice are in accordance with our observations [45]. Interestingly, it was recently described that mouse intra-epithelial T cells express high levels of the GLP-1R by which they regulate the bioavailability of GLP-1 [21]. Whether GLP-1R expression in human T cells plays a similar role is not known. Further studies are required to elucidate the mechanisms and functional roles of GLP-1R signaling in human T cells.

In conclusion, the present study provides first evidence that T-cell activation triggers expression of functional GLP-1R in human T cell subsets, especially in iTreg cells, and that GLP-1R^+^CD4^+^ T cells are found in the dermis of both non-lesional and lesional skin from patients with allergic contact dermatitis. Given the high induction of GLP-1R in human iTreg cells, we hypothesize that GLP-1R^+^ Treg cells play an important role in the anti-inflammatory effects ascribed to GLP-1 in humans.

## Figures and Tables

**Figure 1 cells-11-02587-f001:**
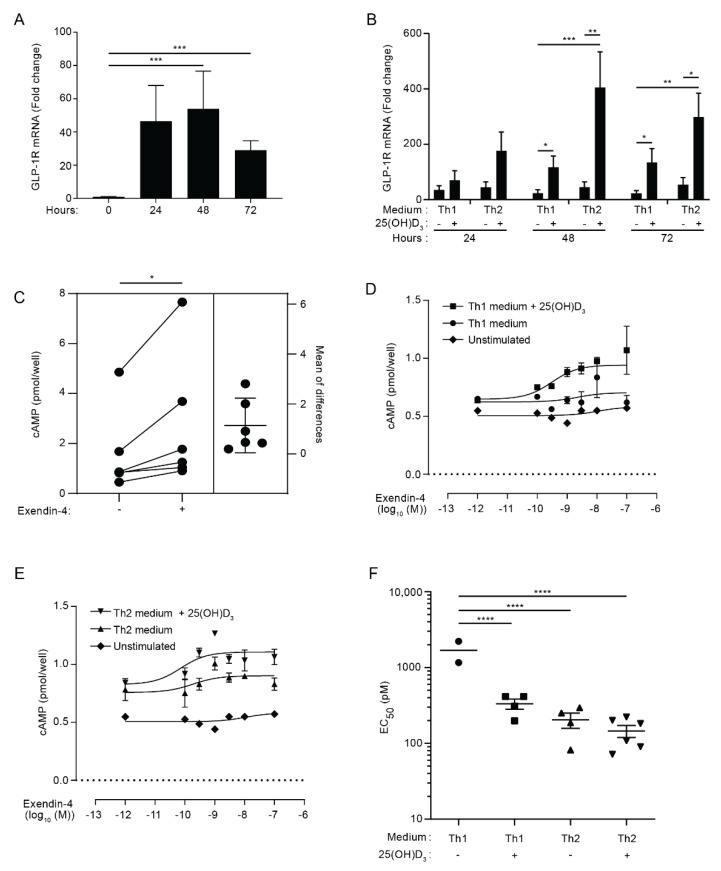
Activated CD4^+^ T cells express functional GLP-1R. (**A**) Relative GLP-1R expression in naïve CD4^+^ T cells (0 h) or CD4^+^ T cells activated for 24, 48, or 72 h. Data are normalized to GLP-1R expression in naïve CD4^+^ T cells (0 h) (mean + SEM, *n* = 6, one-way ANOVA with post-hoc test (Tukey’s), *** *p* ≤ 0.0005). (**B**) Relative GLP-1R expression in CD4^+^ T cells activated for 24, 48, or 72 h in Th1 or Th2 polarizing medium in the absence or presence of 100 nM 25(OH)D_3_ as indicated. Data are normalized to GLP-1R expression in naïve CD4^+^ T cells (mean + SEM, *n* ≥ 6, one-way ANOVA with post-hoc test (Tukey’s) for each time group, * *p* < 0.05, ** *p* < 0.005, *** *p* < 0.0005). (**C**) cAMP in CD4^+^ T cells activated for 60 h in Th2 polarizing medium plus 25(OH)D_3_ (100 nM) and subsequently not treated or treated with exendin-4 (10 nM) for 30 min as indicated (estimation plot, *n* = 6, Student’s *t*-test, (paired, two-tailed), * *p* < 0.05). (**D**,**E**) cAMP accumulation in CD4^+^ T cells activated for 60 h in (**D**) Th1 or (**E**) Th2 medium in the absence or presence of 100 nM 25(OH)D_3_ and treated with increasing amounts of exendin-4 as indicated. Naïve, unstimulated CD4^+^ T cells were included as controls. Representative graphs from three independent experiments with six donors run in duplicates (mean ± SEM). (**F**) EC_50_ values from the nonlinear fits (mean ± SEM, *n* ≥ 4, one-way ANOVA with post-hoc test (Tukey’s), **** *p* < 0.00005).

**Figure 2 cells-11-02587-f002:**
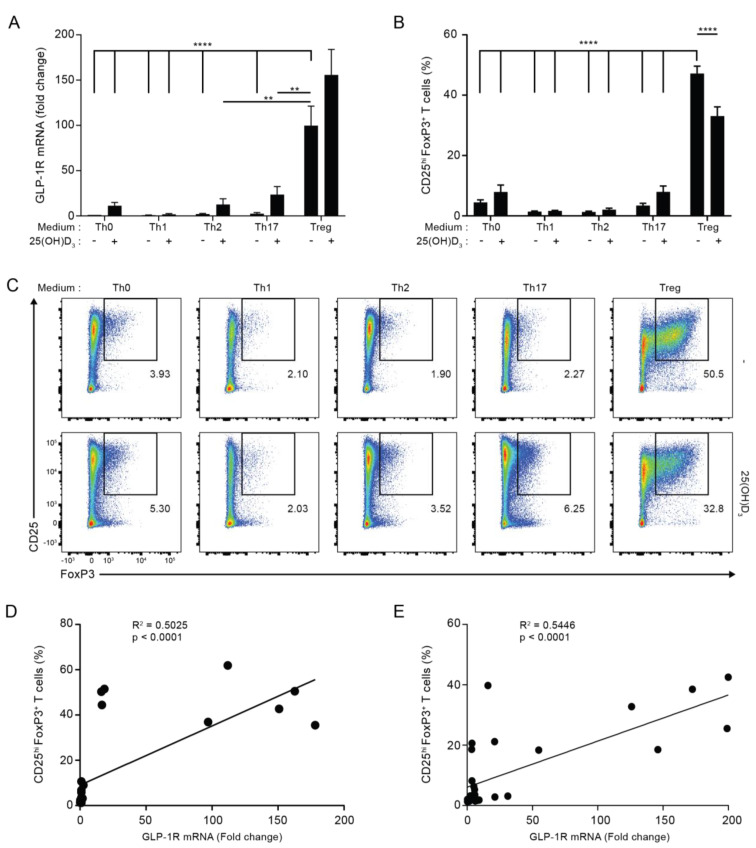
GLP-1R expression directly correlates with the percentage of CD25^+^FoxP3^+^ T cells. (**A**) Relative GLP-1R expression, (**B**) percentage of CD25^+^FoxP3^+^ T cells and (**C**) representative flow cytometric analysis of CD4^+^ T cells activated for 120 h in the indicated polarization medium in the absence or presence of 25(OH)D_3_ (100 nM). GLP-1R expression was normalized to GLP-1R expression in CD4^+^ T cells activated for 120 h in Th0 medium in the absence of 25(OH)D_3_ (mean + SEM, *n* ≥ 4, one-way ANOVA with post-hoc test (Dunnett’s), ** *p* < 0.005, **** *p* < 0.00005. The mean of each column was compared with the mean of the column for Treg minus vitamin D). Data presented in (**B**,**C**) were obtained by gating on live, single lymphocytes as shown in Appendix A. The percentage of CD25^hi^FoxP3^+^CD4^+^ T cells correlated with the expression level of the GLP-1R in CD4^+^ T cells activated for 120 h in the indicated differentiation media in both the absence (**D**) and presence (**E**) of 25(OH)D_3_ (100 nM) as determined by linear regression analysis.

**Figure 3 cells-11-02587-f003:**
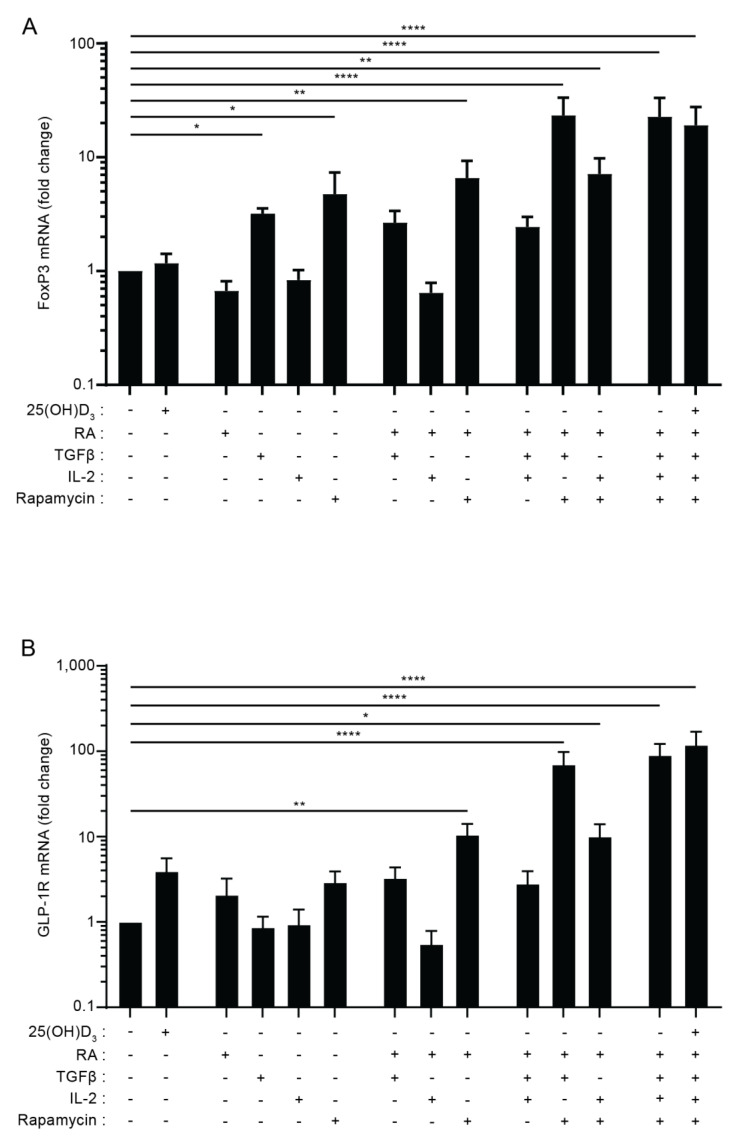
The combination of TGFβ, rapamycin and retinoic acid induces expression of FoxP3 and the GLP-1R in parallel in T cells. Relative FoxP3 (**A**) and GLP-1R (**B**) expression in CD4^+^ T cells activated for 120 h in the presence of the indicated components of the Treg polarizing medium in the absence or presence of 25(OH)D_3_ (100 nM) (RA = retinoic acid, Rapa = Rapamycin). Data were normalized to FoxP3 (**A**) and GLP-1R (**B**) expression in CD4^+^ T cells activated in the absence of any of the components (mean + SEM, *n* = 4, one-way ANOVA with post-hoc test (Dunnett’s), * *p* < 0.05, ** *p* < 0.005, **** *p* < 0.00005. The mean of each column was compared with the mean of the first column).

**Figure 4 cells-11-02587-f004:**
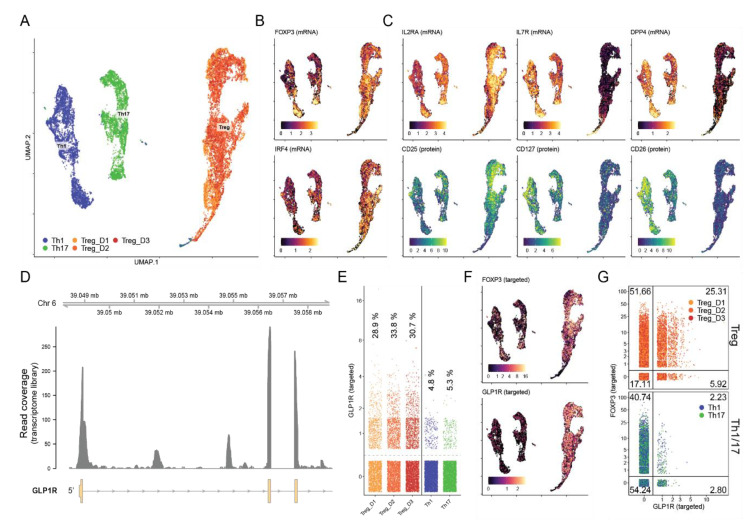
Approximately 30% of iTreg cells express the GLP-1R. (**A**) Visualization of scRNA-seq from CD4^+^ T cells activated in Th1, Th17 or Treg cell-polarizing medium using uniform manifold approximation and projection (UMAP). Cells from three donors—Treg_D1, Treg_D2 and Treg_D3—were included for Treg cells. (**B**,**C**) The iTreg phenotype was confirmed by expression of (**B**) FOXP3 and IRF4 as well as high expression of (**C**) IL2RA (CD25) and low expression of IL7R (CD127) and DPP4 (CD26) at the mRNA (top) and protein (bottom) level using CITE-seq. mRNA expression is shown at the log-normalized UMI count; protein expression was de-noised and normalized using the ‘dsb’ R package. (**D**) Alignment coverage of cDNA reads from the single-cell RNA-seq transcriptome library across the 5′ end of the *GLP1R* locus. Expression of GLP-1R after targeted amplification shown as (**E**) quantification of unique GLP-1R transcripts across donors, (**F**) distribution of GLP-1R^+^ cells across UMAP-space, and (**G**) percentage of cells co-expressing targeted FOXP3 transcripts including percentage of cells within each quadrant shown in the corners.

**Figure 5 cells-11-02587-f005:**
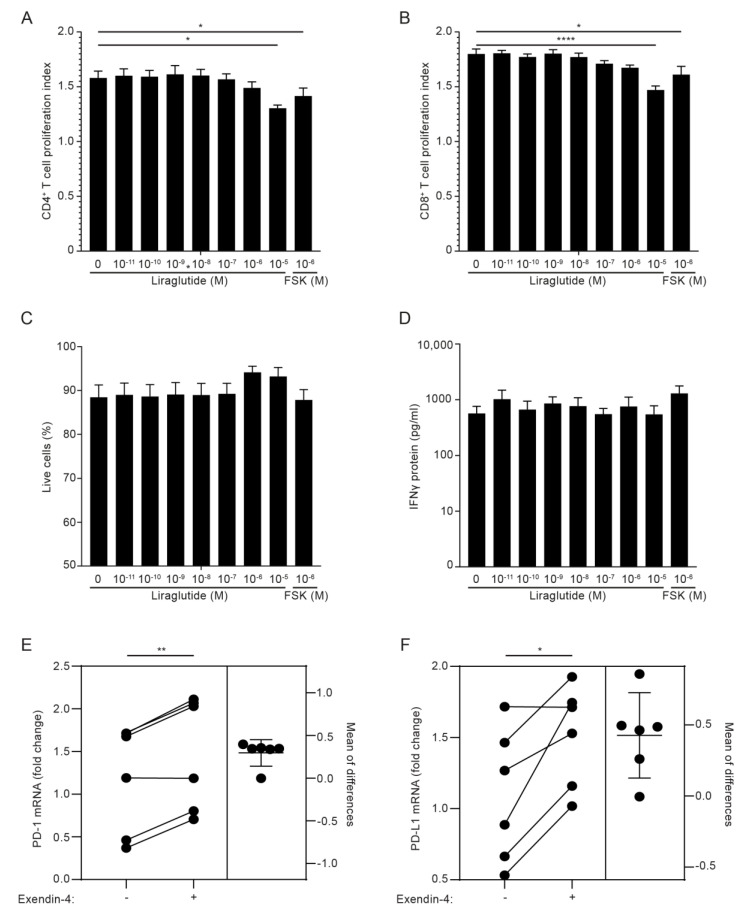
GLP-1RA inhibit proliferation of CD4^+^ and CD8^+^ T cells and increase PD-1 and PD-L1 expression in iTreg cells. (**A**–**D**) Proliferation indexes of CD4^+^ (**A**) and CD8^+^ (**B**) T cells, live cell frequency (**C**) and IFNγ production (**D**) from PBMC cultures stimulated with anti-CD3 mAb for 120 h in the presence of the indicated concentration of the GLP-1RA liraglutide (Lira) or forskolin (FSK) (mean + SEM, *n* = 6, one-way ANOVA with post-hoc test (Dunnett’s), * *p* < 0.05, **** *p* < 0.00005. The mean of each column was compared with the mean of the first column). (**E**,**F**) PD-1 (**E**) and PD-L1 (**F**) expression in CD4^+^ T cells activated for 120 h in Treg polarizing medium and subsequently not treated or treated with exendin-4 (10 nM) for 24 h, as indicated (estimation plot, *n* = 6, Student’s *t*-test, (paired, two-tailed), * *p* < 0.05, ** *p* < 0.005).

**Figure 6 cells-11-02587-f006:**
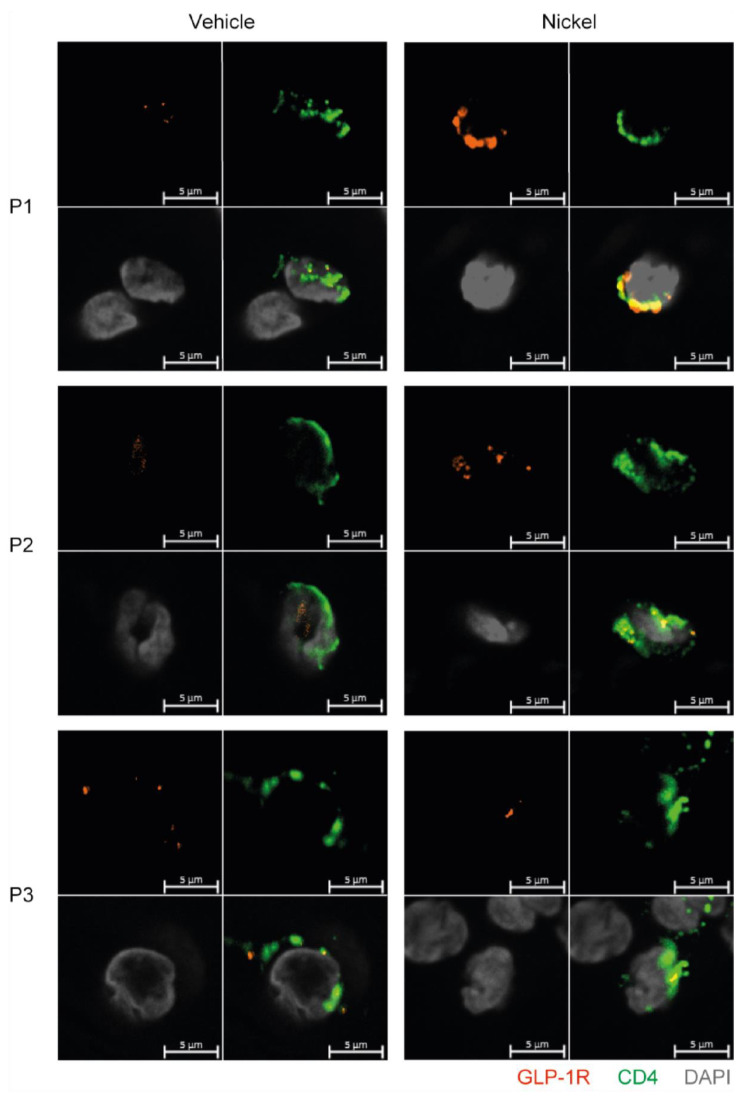
GLP-1R^+^CD4^+^ T cells are found in the skin. First quadrant GLP-1R (red), second quadrant CD4 (green), third quadrant DAPI (grey) and fourth quadrant merged stained fluorescent microscopy images of vehicle-exposed (Vehicle) and nickel-exposed (Nickel) skin from patients with allergic contact dermatitis to nickel. Representative images from three patients (P1, P2, P3).

## Data Availability

Not applicable.

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
