# Peer review of "Induced Human Regulatory T Cells Express the Glucagon-like Peptide-1 Receptor"

_cells, 2022, doi:10.3390/cells11162587_

Round 1

Reviewer 1 Report

Rode et al. demonstrate in this paper that GLP-1 receptors (GLP-1R) are expressed most dominantly by iTreg cells in CD4+ T cells. The results are interesting but confusing. 

1. At first and importantly, since the resolution of Figure 4 is low, I cannot evaluate the data. The authors must make the quality of the figures better. 

2. In some experiments, the authors used exendin-4. But in other experiments, they used liraglutide as GLP-1R agonists (GLP-1RA). Logical explanation is necessary why the authors used these two GLP-1RA in different experiments.

3. The GLP-1R antagonist exendin 9-39 was used to examine the specificity of exendin-4 against GLP-1R. But if exendin 9-39 and exendin-4 can bind to the same membrane proteins other than GLP-1R, inhibition of exendin-4 activity in the presence of exendin 9-39 does not prove the specificity of exendin-4 against GLP-1R. Can the authors make more logical explanation regarding this issue?

4. Interpretation of the results shown in Figures 1 and 2 is not clear. In page 18, lines 17-18, the authors mentioned that "vitamin D did not increase the GLP-1R expression level"...."but rather increased the number of cells expressing the GLP-1R'. To clarify this point, the authors should examine GLP-1R expression in various CD4+ T cells by FACS.

5. In Figure 6, the authors demonstrate the presence of GLP-1R+CD4+ T cells in the skin. But I cannot understand the connection between this analysis and other data (Figures 1-5). Please explain why the authors needed to examine this issue in the text.

Author Response

Thank you so much for your positive and constructive comments.

  1. We completely agree that the resolution of Figure 4 in the merged manuscript is not good. We have consequently included/submitted Figure 4 as a tiff version in high resolution.

  1. After having established that activated T cells express a functional GLP-1R that respond to exendin-4, we wanted to test whether other GLP-1R agonists could affect biological outputs in these cells. We chose to include liraglutide, which is a GLP-1R agonist with a long half-life that recently has become a therapeutic option for patients suffering from diabetes.

  1. Exendin-4 is a high-affinity agonist for the GLP-1R, and interaction between exendin-4 and the GLP-1R increases intracellular cAMP levels to the same extent but with a higher potency than GLP-1. Exendin 9-39 is a potent and specific inhibitor of GLP-1-induced cAMP increase (PMID: 20649595, 10027577, 7851494 and 8405712). To our knowledge, exendin-4 does not induce cAMP through binding to other GPCRs.

  1. We fully agree but unfortunately, no valid antibody specific for the GLP-1R is available for FACS analysis. To account for this experimental short-coming, we instead investigated GLP-1R mRNA expression levels on a single-cell basis, and found iTreg cells to have higher frequency and expression levels compared to differentiated Th1 and Th17 cells. We do however acknowledge that mRNA expression levels might not necessarily correlate to protein levels.

  1. To exclude that GLP-1R expression in T cells was just an in vitro phenomenon, we investigated skin samples from patient with allergic skin dermatitis. This explanation has been included in the revised manuscript.

Reviewer 2 Report

The manuscript written by Rode, et al. described Induced Human Regulatory T Cells Express the Glucagon-Like Peptide-1 Receptor. The authors have detected GLP-1R expression in 29-34% of the FoxP3+CD25+CD127- iTreg cells by using multimodal single-cell CITE- and TCR-sequencing, and they found the presence of GLP-1R+CD4+ T cells in the skin from patients with allergic contact dermatitis. The study have provided the evidence that activated human T cells express the GLP-1R and these foundings demonstrated that T cell activation could enhance the expression of functional GLP-1R in human CD4+ T cells and GLP-1R+ iTreg cells might play a key role in the anti-inflammatory effects. In general, the manuscript is well written and the data is encouraging and important.  but there are some minor points which need to be addressed.

Minor points:

1.     In supplemental Fig.1A, as an antagonist of GPL-1R, why did Exendin 9-39 enhance the production of cAMP in the absence of exendin-4?

2.     The expression level of GLP-1R was similar between Th1 and Th2 cells in the absence of 25(OH)D3 in Fig.1B, why was EC50 significantly different in Fig.1F?

3.     The expression level of GLP-1R was significantly different between Th2 cells in the absence or presence of 25(OH)D3 in Fig.1B, why was there no difference for EC50 between the two groups in Fig.1F?

4.     In Fig.1B and Fig.2A, the difference of the expression of GLP-1R in Th1 and Th2 in the absence or presence of 25(OH)D3 was so different, what is the reason?

5.     The expression level of GLP-1R in resting or activated iTreg cells was much higher than other CD4+ T cell subset such as Th1, Th2, but the expression level of GLP-1R was similar between the resting and activated iTreg cells, how to explain its biological significance?

Author Response

Thank you so much for your positive and constructive comments.

  1. The response elicited by exendin 9-39 as seen in Supplemental Figure 1A is small and not statistically different from the response in untreated cells. A small, not statistically significant response after exendin 9-39 treatment compared to untreated cells is also seen in Supplemental Figure 1B. We believe that these tiny responses reflect experimental uncertainty, but we cannot exclude that exendin 9-39 has a minor partial agonistic activity as reported previously (PMID: 8396143).

  1. As mentioned in paragraph 3.1 “activation resulted in a 30- and 50-fold up-regulation of the GLP-1R expression in CD4+ T cells activated in Th1 and Th2 polarizing medium, respectively, suggesting that T cell differentiation might affect GLP-1R expression (Fig. 1B).” Thus, although not statistically significant in this experiment Th2 cells seem to express more GLP-1R than Th1 cells. As no valid antibody specific for the GLP-1R is available for FACS analysis, we could not determine the GLP-1R expression at the protein level in the single cells but only the mRNA expression in the various sub-populations. We could therefore not determine whether an increased GLP-1R expression as measured in this study reflected i) an increased GLP-1R expression in a given cell, ii) that more cells expressed the GLP-1R but at the same intensity or iii) a combination of the two. We have discussed this and how it would affect the EC50 in the discussion section.

  1. Please see our answer to question two, as this concerns exactly the same problematics.

  1. The differences between Figure 1B and 2A do not reflect different expression levels of the GLP-1R in Th1 and Th2 cells in the two different experiments but just reflect how the data are presented. Thus, in Figure 1B, the GLP-1R expression is shown in Th1 and Th2 cells activated for 24, 48 and 72 h normalized to GLP-1R expression in naïve, unstimulated CD4+ T cells. In contrast, Figure 2A shows the GLP-1R expression in Th0, Th1, Th2, Th17 and Treg cells activated for 120 h normalized to GLP-1R expression in Th0 cells activated for 120 h. The GLP-1R expression levels in Th1 and Th2 cells were similar in the experiments shown in Figure 1B and 2A, respectively.

  1. We do not fully understand the question. We did not investigate the GLP-1R in resting Treg cells but in Treg cells activated in the absence or presence of vitamin D. As mentioned by the reviewer and shown in Figure 2A, activated Treg cells expressed much higher levels of the GLP-1R than activated Th0, Th1, Th2 and T17 cells.

Round 2

Reviewer 1 Report

Still each panel in Figure 4 is too small. I recommend the authors to rearrange the figure so that readers can easily understand the data.